# Interferon beta for the treatment of multiple sclerosis in the Campania Region of Italy: Merging the real-life to routinely collected healthcare data

**Marcello Moccia**[1]*, **Giuseppina Affinito**[2], **Antonio Capacchione**[3], **Roberta Lanzillo**[1], **Antonio Carotenuto**[1], **Emma Montella**[4], **Maria Triassi**[2], **Vincenzo Brescia Morra**[1], **Raffaele Palladino**[2,5]

1 Department of Neuroscience, Reproductive Science and Odontostomatology, Multiple Sclerosis Clinical Care and Research Centre, Federico II University of Naples, Naples, Italy, 2 Department of Public Health, Federico II University of Naples, Naples, Italy, 3 Merck Serono S.p.A (an affiliate of Merck KGaA, Darmstadt, Germany), Rome, Italy, 4 Health Management Office, Federico II University Hospital of Naples, Naples, Italy, 5 Department of Primary Care and Public Health, Imperial College, London, United Kingdom

* moccia.marcello@gmail.com, marcello.moccia@unina.it

**Data Availability Statement:** Data is available upon request to Regional Healthcare Society (So. Re.Sa – www.soresa.it). The authors did not have

## Abstract

### Background

We aim to overcome limitations of previous clinical and population-based studies by merging a clinical registry to routinely-collected healthcare data, and to specifically describe differences in clinical outcomes, healthcare resource utilization and costs between interferon beta formulations for multiple sclerosis (MS).

### Methods

We included 850 patients with MS treated with interferon beta formulations, from 2015 to 2019, seen at the MS Clinical Care and Research Centre (Federico II University of Naples, Italy) and with linkage to routinely-collected healthcare data (prescription data, hospital admissions, outpatient services). We extracted and computed clinical outcomes (relapses, 6-month EDSS progression using a roving EDSS as reference), persistence (time spent on a specific interferon beta formulation), adherence (medication possession ratio (MPR)), healthcare resource utilization and costs (annualized hospitalization rate (AHR), costs for hospital admissions and DMTs). To evaluate differences between interferon beta formulations, we used linear regression (adherence), Poisson regression (AHR), mixed-effect regression (costs), and Cox-regression models (time varying variables); covariates were age, sex, treatment duration, baseline EDSS and adherence.

### Results

Looking at clinical outcomes, rates of relapses and EDSS progression were lower than studies run on previous cohorts; there was no differences in relapse risk between interferon beta formulations. Risk of discontinuation was higher for Betaferon®/Extavia® (HR = 3.28; 95%

any special access privileges that others would not have.

**Funding:** YES This research was partially supported by Merck S.p.A., Rome, Italy, an affiliate of Merck KGaA, Darmstadt, Germany. The funder provided support to the Department of Public Health ("Federico II" University of Naples, Italy), but took no part in the analyses. The final version of this manuscript was approved by the funder, and, then, by all co-authors.

**Competing interests:** Marcello Moccia has received research grants from the ECTRIMS-MAGNIMS, the UK MS Society, and Merck; honoraria from Merck, Roche, and Sanofi-Genzyme; and consultant fees from Veterans' Evaluation Services. Roberta Lanzillo has received honoraria from Biogen, Merck, Novartis, Roche, and Teva. Vincenzo Brescia Morra has received research grants from the Italian MS Society, and Roche, and honoraria from Bayer, Biogen, Merck, Mylan, Novartis, Roche, Sanofi-Genzyme, and Teva. Antonio Capacchione is an employee of Merck Serono S.p.A., Rome, Italy, an affiliate of Merck KGaA, Darmstadt, Germany. Other authors have nothing to disclose.

CI = 2.11, 5.12; p<0.01). Adherence was lower for Betaferon®/Extavia® (Coeff = -0.05; 95%CI = -0.10, -0.01; p = 0.02), and Avonex® (Coeff = -0.06; 95%CI = -0.11, -0.02; p<0.01), when compared with Rebif® and Plegridy® (Coeff = 0.08; 95%CI = 0.01, 0.16; p = 0.02). AHR and costs for MS hospital admissions were higher for Betaferon®/Extavia® (IRR = 2.38; 95%CI = 1.01, 5.55; p = 0.04; Coeff = 14.95; 95%CI = 1.39, 28.51; p = 0.03).

## Conclusions

We have showed the feasibility of merging routinely-collected healthcare data to a clinical registry for future MS research, and have confirmed interferon beta formulations play an important role in the management of MS, with positive clinical outcomes. Differences between interferon beta formulations are mostly driven by adherence and healthcare resource utilization.

## Introduction

In the past decades, several injectable, oral and monoclonal antibody disease modifying treatments (DMTs) have become available for multiple sclerosis (MS) [1]. However, DMTs have been rarely compared directly in relation to clinical and healthcare outcomes. On the one hand, MS registries include clinical and treatment data, but are at risk of patient selection (e.g., inclusion of patients and clinical variables only from participating centres), and follow-up (e.g., variable follow-up duration, with patients doing poorly being most likely to be lost to follow-up) [2,3]. On the contrary, datasets based on routinely-collected healthcare data provide detailed healthcare resource utilization with high external validity, in the long-term and on fully representative populations, but lack of clinical data [4].

In our previous studies, we have differentiated interferon beta formulations for the treatment of MS using our clinical registry [5], and, separately, using routinely-collected healthcare data of the Campania Region of Italy [4,6], and showed that Rebif® might be characterized by better efficacy and healthcare utilization profile, when compared with other formulations. Hereby, we aim to overcome limitations of our previous studies by merging real-world clinical data to routinely collected healthcare data, to describe differences in clinical outcomes, healthcare resource utilization and costs between interferon beta formulations.

## Methods

### Study design and population

The present observational cohort study is a retrospective analysis of prospectively collected data on people living with MS attending the MS Clinical Care and Research Centre at the Federico II University of Naples, which were linked to routinely-collected healthcare data (prescription data, hospital admissions, outpatient services).

Study population was defined considering the following inclusion criteria: 1) diagnosis of MS and clinical follow-up at the MS Clinical Care and Research Centre (Federico II University of Naples); 2) 2015–2019 year range; 3) interferon beta prescription and utilization for at least 3 months. The MS population of the MS Clinical Care and Research Centre at the Federico II University of Naples is thought to be representative of the MS population of the Campania Region [6,7]. Exclusion criteria were: 1) age < 18 years; 2) incomplete clinical records.

Anonymisation was performed using the same algorithm on clinical registry and routinely-collected healthcare data to allow data linkage. Data extraction and linkage was approved by the Federico II Ethics Committee (355/19). All patients signed informed consent authorising the use of anonymised and aggregated data collected routinely as part of the clinical practice, in line with data protection regulation (GDPR EU2016/679). The study was performed in accordance with good clinical practice and Declaration of Helsinki.

## Clinical outcomes

Clinical outcomes were extracted from the clinical registry and were referred to each individual treatment period. During follow-up, patients were evaluated every 3 months, or on the occurrence of a clinical relapse, by an Expanded Disability Status Scale (EDSS) qualified neurologist. The following major clinical outcomes were extracted: occurrence of clinical relapse, time from baseline to the first relapse (time to first relapse), annualized relapse rate (ARR), EDSS progression, and time to EDSS progression (confirmed after 6 months, using a roving EDSS as reference) [8]. Disease duration was estimated as the time between reported clinical onset and baseline.

## Persistence and adherence

DMT supply was obtained from electronic records of pharmacy services. Persistence was measured as the time spent on a specific DMT (related to each individual treatment period) [9]. Medication possession ratio (MPR) was calculated as an indirect measure of adherence (MPR = (medication supply obtained during follow-up period/medication supply expected during follow-up period)*100) [10].

## Healthcare resource utilization and costs

As from our previous paper [6], healthcare resource utilization was extracted from Campania Region datasets (i.e., hospital discharge records, regional prescribing database, and outpatient services). Based on the number of inpatient hospital admissions, we computed the annualized hospitalization rate (AHR). Healthcare costs were derived from the Regional registry for corresponding healthcare resource utilisation [4], and were inflated to the most recent values (2019), in order to avoid variations in price per unit of service through different years, and were reported on a monthly basis. For patients with hospital discharge records, we computed the Charlson Comorbidity Index [11].

## Statistics

Descriptive statistics were performed as appropriate considering each variable distribution. To evaluate differences in study variables between interferon beta formulations, we used mixed-effect linear regression models (for adherence and costs), Poisson regression models (for ARR and AHR), and Cox-regression models (for time varying variables, such as time to DMT discontinuation, first relapse, EDSS progression). Rebif® was used as reference in the statistical models. Covariates were age, sex, treatment duration, baseline EDSS and adherence (MPR). Results were presented as coefficients (Coeff), incidence rate ratio (IRR), hazard ratios (HR), 95% confidence interval (95%CI), and p-values, as appropriate. Results were considered statistically significant if $p < 0.05$. Stata 15.0 was used for data processing and analysis.

## Results

We included 850 patients with MS treated with interferon beta formulations, for overall 887 individual treatment periods (with some patients being treated with different interferon beta formulations during the study period). Patient disposition flow diagram is presented in **Fig 1**. Demographics, clinical features, persistence, adherence, healthcare resource utilization and costs are reported in **Table 1**.

ARR was lower for Avonex® (IRR = 0.61; 95%CI = 0.40, 0.93; p = 0.02), while there was no significant difference between Rebif®, Betaferon®/Extavia® (IRR = 0.71; 95%CI = 0.46, 1.10; p = 0.12), and Plegridy® (IRR = 0.26; 95%CI = 0.06, 0.93; p = 0.06). There was no significant difference in relapse risk (time to first relapse) between Rebif®, Avonex® (HR = 0.40; 95%CI = 0.15, 1.06; p = 0.06), Betaferon®/Extavia® (HR = 0.67; 95%CI = 0.28, 1.62; p = 0.38), and Plegridy® (HR = 0.57; 95%CI = 0.13, 2.38; p = 0.44). Risk of roving EDSS progression was lower for Avonex® (HR = 0.29; 95%CI = 0.11, 0.77; p = 0.01), while there was no significant difference between Rebif®, Betaferon®/Extavia® (HR = 0.90; 95%CI = 0.41, 1.96; p = 0.79), and Plegridy® (HR = 0.72; 95%CI = 0.31, 1.69; p = 0.45).

Risk of discontinuation was 3.3-fold greater for Betaferon®/Extavia® (HR = 3.28; 95% CI = 2.11, 5.12; p<0.01), while there was no significant difference between Rebif®, Avonex® (HR = 0.92; 95%CI = 0.66, 1.29; p = 0.63) and Plegridy® (HR = 1.24; 95%CI = 0.88, 1.75; p = 0.21).

Adherence was 5% lower for Betaferon®/Extavia® (Coeff = -0.05; 95%CI = -0.10, -0.01; p = 0.02), 6% lower for Avonex® (Coeff = -0.06; 95%CI = -0.11, -0.02; p<0.01), and 8% higher for Plegridy® (Coeff = 0.08; 95%CI = 0.01, 0.16; p = 0.02), as compared with patients taking Rebif®.

There were 35 hospital admissions during the study period. AHR was greater for Betaferon®/Extavia® (IRR = 2.38; 95%CI = 1.01, 5.55; p = 0.04), while there was no significant difference between Rebif®, Avonex® (IRR = 1.54; 95%CI = 0.56, 4.19; p = 0.39), and Plegridy® (IRR = 1.61; 95%CI = 0.19, 13.20; p = 0.65). Costs for hospital admissions were higher for Plegridy® (Coeff = 22.98; 95%CI = 9.65, 36.32; p<0.01), while there was no significant difference between Rebif®, Avonex® (Coeff = 3.41; 95%CI = -7.57, 14.41; p = 0.54), and Betaferon®/Extavia® (Coeff = 0.83; 95%CI = -11.64, 13.31; p = 0.89). Costs for MS hospital admissions were higher for Betaferon®/Extavia® (Coeff = 14.95; 95%CI = 1.39, 28.51; p = 0.03), while there was no significant difference between Rebif®, Plegridy® (Coeff = 2.88;

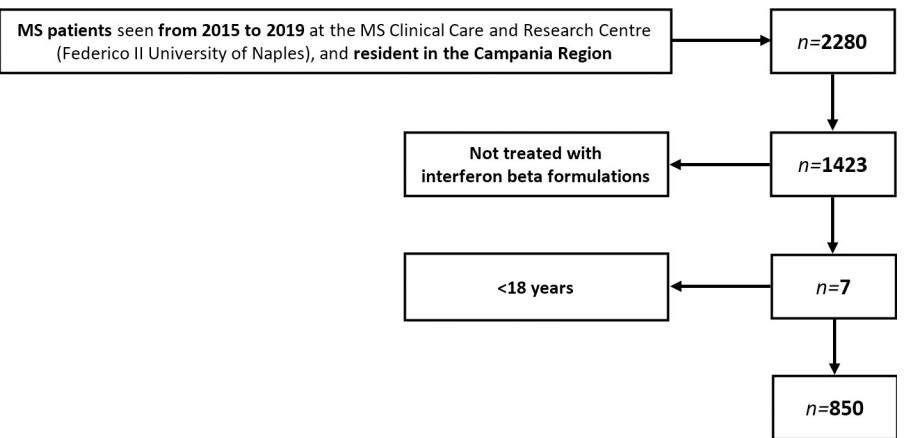

**Fig 1. Patient disposition flow diagram.**

**Table 1. Demographics, clinical features, persistence, adherence, healthcare resource utilization and costs.**

| | | | Rebif® | Avonex® | Plegridy® | Betaferon®/Extavia® |
|---|---|---|---|---|---|---|
| Patients, *n* | | | 361 | 231 | 60 | 198 |
| Individual treatment periods, *n* | | | 382 | 238 | 60 | 207 |
| Females, *n (%)* | | | 250 (69.2%) | 173 (74.9%) | 47 (78.3%) | 132 (66.7%) |
| Age, *years* | | | 35.7±10.4 | 39.4±10.5 | 39.4±9.6 | 42.3±11.9 |
| Charlson comorbidity index* | | *0* | 368 | 227 | 60 | 194 |
| | | *1–2* | 2 | 6 | 0 | 8 |
| | | *≥3* | 0 | 0 | 0 | 0 |
| Disease duration, *years* | | | 2.8±1.8 | 3.5±1.9 | 1.9±1.2 | 2.9±1.7 |
| EDSS at baseline | | | 3.01±1.21 | 2.95±0.99 | 2.51±1.11 | 4.22±1.54 |
| Relapse occurrence, *n* | | | 81 | 37 | 3 | 30 |
| ARR | | | 0.16±0.54 | 0.11±0.41 | 0.06±0.31 | 0.09±0.37 |
| Roving EDSS progression, n | | | 149 | 93 | 0 | 112 |
| Adherence (MPR) | | | 0.84±0.29 | 0.81±0.31 | 0.92±0.32 | 0.80±0.27 |
| Treatment discontinuation, *n* | | | 91 (23.8%) | 67 (28.1%) | 30 (50.0%) | 65 (31.4%) |
| Time to discontinuation, *years* | | | 2.87±1.86 | 3.52±1.90 | 1.95±1.21 | 2.90±1.72 |
| AHR | | | 0.01±0.07 | 0.02±0.21 | 0.01±0.07 | 0.05±0.27 |
| Hospital admission costs, *EUR* | | | 36.98±41.00 | 40.12±70.98 | 45.59±32.32 | 47.82±97.90 |
| MS hospital admission costs, *EUR* | | | 34.08±30.70 | 32.67±40.89 | 45.59±32.32 | 41.26±86.87 |
| DMT costs, *EUR* | | | 886.74±275.26 | 701.26±218.72 | 796.78±258.01 | 423.29±134.33 |

*For patients with hospital discharge records.

95%CI = -6.70, 12.46; p = 0.55), and Avonex® (Coeff = -3.37; 95%CI = -12.07, 5.32; p = 0.44). Costs for DMTs were lower for Avonex® (Coeff = -157.29; 95%CI = -182.28, -132.29; p<0.01), Plegridy® (Coeff = -131.28; 95%CI = -173.60, -88.96; p<0.01), and Betaferon®/Extavia® (Coeff = -452.80; 95%CI = -480.15, -425.46; p<0.01), as compared with patients taking Rebif®.

## Discussion

In the present study, we have confirmed our previous clinical and population-based results on the use of interferon beta formulations [4,5,9,12], and have showed the feasibility of merging routinely-collected healthcare data and clinical registry for future MS research.

One third of MS patients have received at least one prescription of interferon beta from 2015 to 2019, with Rebif® being the preferred interferon beta formulation, especially in young patients [4]. We have confirmed that adherence is kept at optimal levels in our centre (overall above 80%) [4,12], with higher rates in Rebif® and Plegridy®, when compared with Betaferon®/Extavia® and Avonex®. Also, MS patients remained on interferon beta treatment for 2–3 years, with higher discontinuation rates for Betaferon®/Extavia®, when compared with Rebif®, Avonex®, and Plegridy®. Costs were mainly driven by the use of DMTs, though some interferon beta formulations (e.g., Rebif®, Avonex®) are associated with reduced rates of hospital admissions and related costs. Looking at clinical outcomes, rates of relapses and disability progression (estimated using a roving EDSS as reference) were lower than studies run on previous cohorts [5], possibly also as a consequence of new diagnostic criteria [13], with difficulties in finding and interpreting statistical differences. For instance, we found no differences in time to the first relapse, but in overall ARR, suggesting these differences are a consequence of swtiching timeliness, with some patients not being switched to

more effective DMTs after the first relapse and, thus, accumulating additional relapses. Similarly, differences in rates of disability progression might be biased by the available follow-up to establish sustained progression, which is possibly further increased by the use of a roving EDSS as reference [8].

Our study suffers from different limitations, mostly arising from the single centre design and differences in baseline characteristics, that we tried to mitigate by using covariates in the statistical models. However, we have showed the feasibility of combining routinely-collected healthcare data to clinical register, for future MS research. We confirmed that interferon beta formulations play an important role in the management of MS, and are overall associated with positive clinical outcomes in the mid-term. Differences between interferon beta formulations are mostly driven by adherence and healthcare resource utilization.

## Acknowledgments

Marcello Moccia has received research grants from the ECTRIMS-MAGNIMS, the UK MS Society, and Merck; honoraria from Merck, Roche, and Sanofi-Genzyme; and consultant fees from Veterans' Evaluation Services. Roberta Lanzillo has received honoraria from Biogen, Merck, Novartis, Roche, and Teva. Vincenzo Brescia Morra has received research grants from the Italian MS Society, and Roche, and honoraria from Bayer, Biogen, Merck, Mylan, Novartis, Roche, Sanofi-Genzyme, and Teva. Antonio Capacchione is an employee of Merck Serono S.p. A., Rome, Italy, an affiliate of Merck KGaA, Darmstadt, Germany. Other authors have nothing to disclose.

## Author Contributions

**Conceptualization:** Marcello Moccia, Antonio Capacchione, Maria Triassi, Vincenzo Brescia Morra, Raffaele Palladino.

**Data curation:** Marcello Moccia, Giuseppina Affinito, Roberta Lanzillo, Antonio Carotenuto, Emma Montella, Vincenzo Brescia Morra, Raffaele Palladino.

**Formal analysis:** Giuseppina Affinito, Roberta Lanzillo, Antonio Carotenuto, Emma Montella.

**Funding acquisition:** Antonio Capacchione.

**Investigation:** Marcello Moccia, Giuseppina Affinito, Antonio Capacchione, Roberta Lanzillo, Antonio Carotenuto, Emma Montella, Maria Triassi, Vincenzo Brescia Morra, Raffaele Palladino.

**Methodology:** Marcello Moccia, Giuseppina Affinito, Antonio Capacchione, Antonio Carotenuto, Emma Montella, Raffaele Palladino.

**Project administration:** Marcello Moccia, Antonio Capacchione, Maria Triassi.

**Resources:** Roberta Lanzillo, Vincenzo Brescia Morra.

**Software:** Giuseppina Affinito, Roberta Lanzillo, Antonio Carotenuto.

**Supervision:** Marcello Moccia, Maria Triassi, Vincenzo Brescia Morra, Raffaele Palladino.

**Validation:** Antonio Carotenuto, Emma Montella.

**Visualization:** Antonio Carotenuto.

**Writing – original draft:** Marcello Moccia, Antonio Capacchione, Maria Triassi, Vincenzo Brescia Morra, Raffaele Palladino.

**Writing – review & editing:** Giuseppina Affinito, Roberta Lanzillo, Antonio Carotenuto, Emma Montella.

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
