## [Decision Letter · Decision Letter 0]

9 Aug 2021

PONE-D-21-18561

Interferon beta for the treatment of multiple sclerosis in the Campania Region of Italy: merging the real-life to routinely collected healthcare data

PLOS ONE

Thank you for submitting your manuscript to PLOS ONE. After careful consideration, we feel that it has merit but does not fully meet PLOS ONE’s publication criteria as it currently stands. Therefore, we invite you to submit a revised version of the manuscript that addresses the points raised during the review process.

Please submit your revised manuscript by August 31th. If you will need more time than this to complete your revisions, please reply to this message or contact the journal office at plosone@plos.org. Please include the following items when submitting your revised manuscript:

We look forward to receiving your revised manuscript.

Kind regards,

Luigi Lavorgna

Academic Editor

PLOS ONE

Journal Requirements:

3. We note that you have stated that you will provide repository information for your data at acceptance. Should your manuscript be accepted for publication, we will hold it until you provide the relevant accession numbers or DOIs necessary to access your data. If you wish to make changes to your Data Availability statement, please describe these changes in your cover letter and we will update your Data Availability statement to reflect the information you provide

4. We noticed you have some minor occurrence of overlapping text with the following previous publication, which needs to be addressed:

- https://bmchealthservres.biomedcentral.com/articles/10.1186/s12913-020-05664-x

In your revision ensure you cite all your sources (including your own works), and quote or rephrase any duplicated text outside the methods section. Further consideration is dependent on these concerns being addressed

Reviewers' comments:

Reviewer's Responses to Questions

**Comments to the Author**

1. Is the manuscript technically sound, and do the data support the conclusions?

Reviewer #1: Yes

Reviewer #2: Yes

2. Has the statistical analysis been performed appropriately and rigorously? 

Reviewer #1: Yes

Reviewer #2: Yes

3. Have the authors made all data underlying the findings in their manuscript fully available?

Reviewer #1: Yes

Reviewer #2: Yes

4. Is the manuscript presented in an intelligible fashion and written in standard English?

Reviewer #1: Yes

Reviewer #2: Yes

5. Review Comments to the Author

Reviewer #1: In this interesting work, the authors found that differences between interferon beta formulations in Campania region were related to adherence and healthcare resource utilization, by merging routinely-collected healthcare data to a clinical registry. They showed that this innovative approach for data analysis is feasible and can therefore be largely used in future research.

The work is clear and well written, I only have some minor comments:

- In this cohort of patients, the risk of discontinuation was the highest in patients treated with Betaferon®/Extavia®, which were those with higher EDSS at baseline. Can the authors provide any reassurance that the worse disability at the beginning of the disease did not impact the analysis?

- Have the authors explored whether different interferon beta formulations influence the risk of time to switch to a second line MS treatment and time to conversion to secondary progressive disease course, if data after 2019 were available?

- Disease duration of MS patients at baseline should be added to Table 1.

Reviewer #2: In this article Moccia and colleagues describe differences in clinical outcomes, healthcare resource utilization and costs between interferon beta formulations for MS, merging a clinical registry to routinely-collected healthcare data. Methods are sound, the statistical analysis has been performed appropriately and rigorously.

There is only a minor concern. Regarding the sentence: “Looking at clinical outcomes, rates of relapses and EDSS progression were lower than studies run on older cohorts.” not is clear if the authors mean older age cohorts or previous cohort. It should be clarified. Furthermore, in the opinion of the authors, the introduction of the new diagnostic criteria in the recent years, could have influenced the results of the study and the differences with older age cohorts? (PMID: 30419509).

6. PLOS authors have the option to publish the peer review history of their article (what does this mean?). If published, this will include your full peer review and any attached files.

Reviewer #1: No

Reviewer #2: No

---

## [Author Response · Author response to Decision Letter 0]

13 Sep 2021

Reviewer #1

In this interesting work, the authors found that differences between interferon beta formulations in Campania region were related to adherence and healthcare resource utilization, by merging routinely-collected healthcare data to a clinical registry. They showed that this innovative approach for data analysis is feasible and can therefore be largely used in future research. The work is clear and well written, I only have some minor comments.

We thank the reviewer for his/her positive feedback.

In this cohort of patients, the risk of discontinuation was the highest in patients treated with Betaferon®/Extavia®, which were those with higher EDSS at baseline. Can the authors provide any reassurance that the worse disability at the beginning of the disease did not impact the analysis?

We have adjusted all statistical models for baseline EDSS, as reported in the statistical methods. We have now also highlighted this in the limitations’ section of the Discussion:

“Our study suffers from different limitations, mostly arising from …differences in baseline characteristics, that we tried to mitigate by using covariates in the statistical model”.

Have the authors explored whether different interferon beta formulations influence the risk of time to switch to a second line MS treatment and time to conversion to secondary progressive disease course, if data after 2019 were available?

Unfortunately data after 2019 were not available and, as also suggested by the Reviewer, we decided not to evaluate conversion to secondary progressive disease course.

On the contrary, we have now evaluated the time to second-line DMT as shown in the table below.

Switch to second line DMT Coeff. P.value 95% Conf. Interval

Rebif® Reference

Avonex® -0.589 0.006 -1.007 -0.172

Plegridy® 0.233 0.475 -0.406 0.872

Betaferon®/Extavia® 0.057 0.770 -0.326 0.441

Overall, results mirror what we have already found in relation to ARR, and, thus, do not seem to add much to the paper. As such, we have decided not to include this analysis in the main body of the manuscript which actually focuses on clinical outcomes (relapses and disability).

Disease duration of MS patients at baseline should be added to Table 1.

We have now added disease duration to Table 1.

Reviewer #2

In this article Moccia and colleagues describe differences in clinical outcomes, healthcare resource utilization and costs between interferon beta formulations for MS, merging a clinical registry to routinely-collected healthcare data. Methods are sound, the statistical analysis has been performed appropriately and rigorously.

We thank the reviewer for his/her positive feedback.

There is only a minor concern. Regarding the sentence: “Looking at clinical outcomes, rates of relapses and EDSS progression were lower than studies run on older cohorts.” not is clear if the authors mean older age cohorts or previous cohort. It should be clarified.

As suggested, we have revised to “previous cohort”. 

Furthermore, in the opinion of the authors, the introduction of the new diagnostic criteria in the recent years, could have influenced the results of the study and the differences with older age cohorts? (PMID: 30419509).

We thank the Reviewer for suggesting this, and have now revised the following sentences in the Discussion (along with reference):

“Looking at clinical outcomes, rates of relapses and disability progression (estimated using a roving EDSS as reference) were lower than studies run on previous cohorts, possibly also as a consequence of new diagnostic criteria”

---

## [Editor Report · Decision Letter 1]

16 Sep 2021

Interferon beta for the treatment of multiple sclerosis in the Campania Region of Italy: merging the real-life to routinely collected healthcare data

PONE-D-21-18561R1

We’re pleased to inform you that your manuscript has been judged scientifically suitable for publication and will be formally accepted for publication once it meets all outstanding technical requirements.

Kind regards,

Luigi Lavorgna

Academic Editor

PLOS ONE
---

## [Editor Report · Acceptance letter]

20 Sep 2021

PONE-D-21-18561R1 

Interferon beta for the treatment of multiple sclerosis in the Campania Region of Italy: merging the real-life to routinely collected healthcare data 

Dear Dr. Moccia:

I'm pleased to inform you that your manuscript has been deemed suitable for publication in PLOS ONE. Congratulations! Your manuscript is now with our production department. 

Kind regards, 

on behalf of

Dr. Luigi Lavorgna 

Academic Editor

PLOS ONE